# Captive Animal Behavior Study by Video Analysis

**DOI:** 10.3390/s23187928

**Published:** 2023-09-16

**Authors:** Florin Rotaru, Silviu-Ioan Bejinariu, Hariton-Nicolae Costin, Ramona Luca, Cristina Diana Niţă

**Affiliations:** Institute of Computer Science, Romanian Academy Iasi Branch, T. Codrescu Str., 2, 700481 Iaşi, Romania; florin.rotaru@iit.academiaromana-is.ro (F.R.); silviu.bejinariu@iit.academiaromana-is.ro (S.-I.B.); ramona.luca@iit.academiaromana-is.ro (R.L.); cristina.nita@iit.academiaromana-is.ro (C.D.N.)

**Keywords:** video tracking, trajectory analysis, animal behavior, stress detection

## Abstract

Three video analysis-based applications for the study of captive animal behavior are presented. The aim of the first one is to provide certain parameters to assess drug efficiency by analyzing the movement of a rat. The scene is a three-chamber plastic box. First, the rat can move only in the middle room. The rat’s head pose is the first parameter needed. Secondly, the rodent could walk in all three compartments. The entry number in each area and visit duration are the other indicators used in the final evaluation. The second application is related to a neuroscience experiment. Besides the electroencephalographic (EEG) signals yielded by a radio frequency link from a headset mounted on a monkey, the head placement is a useful source of information for reliable analysis, as well as its orientation. Finally, a fusion method to construct the displacement of a panda bear in a cage and the corresponding motion analysis to recognize its stress states are shown. The arena is a zoological garden that imitates the native environment of a panda bear. This surrounding is monitored by means of four video cameras. We have applied the following stages: (a) panda detection for every video camera; (b) panda path construction from all routes; and (c) panda way filtering and analysis.

## 1. Introduction

The automatic analysis of animal behavior via video following has become a significant matter in neurosciences. To evaluate the effects of a gamut of factors on humans, such as drug and medicine influence, strain differences, or genetic mutations, medical research uses laboratory animals, mainly rodents. For instance, to evaluate the efficiency of a new drug to cure a specific disease, chemicals are administrated to rodents to induce specific symptoms. General or episodic facts, such as rearing, grooming, chewing, or movement in certain surroundings, have to be analyzed before and after drug administration. Laboratory beings are also utilized in neurology and psychiatry research to increase the basic knowledge of the nervous system and the anatomy and physiology of the brain. This cognition leads to better comprehension and therefore to better care of neurological and psychic illnesses. To substantiate the behavior of the human brain, monkeys are much more suited than rodents. For this type of experiment, the analysis of the monkeys’ behavior under specific circumstances is the most efficient way to realize the connections between individual or certain neuronal structures and complex cognitive functions. Another specific field of following animal behavior is intended to identify the motion stereotypes showing stress episodes of animals in zoos. A link between the stress state and some outside agents as visitors or other cumbersome facts that could provoke it has also to be established.

To objectively measure the relevant behaviors, many automated video monitoring and motion investigation techniques have been studied. There are proposals where the specificity of the scene imposes very particular tracking and trajectory analysis algorithms [1,2,3,4,5]. A recent review of the currently available open-source tools for animal behavioral video analysis is presented in [6]. Another important type of the proposed organism motion monitoring applications uses Kalman filter-based methods [7,8,9,10]. Other articles present animal activity tracking by means of AI (deep learning) or the optical flow method [11,12,13,14,15,16,17,18]. Two methods to localize rodents and analyze their motion are presented in [1,2]. The first one is a two-step approach to identify rodents in specific surroundings with few restraints. First, a sliding window method based on three traits follows the rodent and calculates its raw place. Next, the edge map and a system of pulses are employed to set the borders of the watching window close to the contour of the rodent. The other approach proposes an algorithm coming from the CONDENSATION method, which uses a smaller average number of provided tracking particles. This technique decreases the computational load of the initial algorithm while keeping the same detection accuracy. Analysis of the synthesized trajectory identified three rodent behaviors (exploring, rearing, and freezing). An improved version of motion analysis was later presented in [3]. Motion History Images and a Multiple Classifier System were used to recognize the three comportments. In every decision stage, the classifier with the best decision was taken into account. The presented method is invariant to the rodent and background color. It provides an accuracy of 87%. 

To assess the animal activity related to a specific pathology, some techniques of trajectory estimation are presented in [4]: (a) two differential techniques by means of background subtraction; (b) entropy thresholding proceeding by using a global and local threshold; and (c) color matching with a computed template integrating a selected spectrum of colors. The authors deduced that all used procedures offered similar results for small rodent activities, but these techniques were aborted when great motions were analyzed. 

A method to assess the fast and periodic behavior of a mouse by extracting the scratching patterns has been shown in [5]. Motion is identified by a series of operations in the consecutive binarized frames differences: noise reduction by erosion; nonzero pixels counting; pulse detection; long-term pulse filtering; pulse processing for scratching quantification; and scratching time quantification. The outcome is interpreted as an objective behavioral evaluation of atopic dermatitis, which can help to obtain new drugs for this disease. 

In this paragraph, some tracking organism applications using Kalman filter-based techniques are remembered. An available tracking method and program software for animal activity named ToxTrac is described in [7]. It was developed for high-speed following of insects, fish, and rodents. First, it identifies gray mobile animals on a bright homogenous background. Next, it uses a second-order Kalman filter to assess target paths. 

‘Bio-Sense’, described in [8], is a computing program that ensures object detection and then employs an adaptive Kalman filter to model the motion dynamics of different animals. 

The paper [9] presents a simple following algorithm that accommodates the spatially closest objects of interest between successive video frames. It approaches only objects marked as animals in a previous stage. 

An accommodation of the Kalman filter is used to estimate the position of animals that are hidden by other objects. For these methods, the main supposition is the motion linearity. The Kalman filter method is hard to develop when the animal route is covered. There were deployed methods to defeat this shortcoming. For example, in [10], a mixture of mean shift and particle–Kalman filter methods is shown. Mean shift performs as the main follower when the object is not covered. When occlusions appear, the particle–Kalman filter becomes the leader of the procedure. But for some experimental research, as in our application, the movement linearity is not fulfilled due to the sudden changes in speed and movement direction. 

A review concerning deep learning tools for the evaluation of animal activity in neuroscience was issued in [11]. The study reveals that even though pose estimation based on deep learning can now much more easily be measured, there still remain some challenges to be solved. Performances of pose estimation approaches have to be improved for situations with contexts of multi-animal/human or/and occlusions. Also, it is mentioned that right now, it is hard to train networks to generalize to out-of-domain scenarios. 

A video analysis of the animal posture evaluation framework is presented in [12]. It sums a deep learning elastic exemplary to approach the fluctuation in the animal corpus profile and an optical flow method to integrate temporal information from different film sequences. The proposed application was evaluated using datasets of four different lab animal species (mouse, fruit fly, zebrafish, and monkey). The authors claim that their system provides the best prediction accuracy among current deep learning-based approaches. 

To identify multiple animals in areas with many occlusions and crossings, the following technique is described in [13]. The main idea is to link trajectory parts by means of their histogram and Hu moments. 

The authors of [14] demonstrate the application of deep learning for the detection, tracking, activity, and behavior determination of red foxes during an experimental study. The detector was trained to detect red foxes and three body postures (‘lying’, ‘sitting’, ‘standing’). The trained algorithm has a mean average precision of 99.91%. 

One deep learning method uses certain deep neural networks for panda bear face detection, segmentation, alignment, and identification [15]. The authors assert that the method is credited with 96.27% accuracy in panda identification. 

The authors of [16] proposed an unsupervised probabilistic deep learning framework that identifies behavioral structures from deep variational embeddings of animal motion. In [17], the authors claim to demonstrate, for the first time, the effective application of a deep learning-based head pose estimation model trained and tested in a non-restraint setup with a monkey taking part in a cognitive task in their home area. Another deep learning framework for non-invasive behavior monitoring of individual animals is proposed in [18]. Video data are processed to continuously determine the position of a polar bear within the area. The analysis of the bear’s trajectories describing his travel patterns is presented through a graphical interface. The authors show the framework can localize and identify individual polar bears with a rate of 86.4%.

The remainder of this paper is structured as follows: The following chapter presents the experiment environments and the animal identification methods applied in three applications: (a) a laboratory rat tracking and trajectory analysis technique to provide certain movement parameters used to assess a specific drug efficiency; (b) a monkey tracking and head orientation computation method that provides information related to a neuroscience experiment; (c) a zoo panda bear motion analysis to identify stress states due to guests and the noise produced by airplanes when using a neighboring airport. A resume of laboratory rat tracking and a trajectory analysis system is presented in Section 3. The application is proposed in [19,20]. Section 4 describes, in the same manner, the laboratory monkey tracking and trajectory analysis system whose detailed presentation was the subject of [21]. The detailed proposal of an application for zoo panda stress state identification is presented in Section 5. The last application is briefly described in [22]. At the end of each of the last three sections, the results and conclusions are presented.

## 2. Experimental Environments and Localization Methods

A plastic box divided into three rooms by means of two walls represented the environment of the first presented application. A cage was placed in the left room and another in the right room. Each video showing the experiments began with some frames displaying a rat placed in the middle domain. The scene is seen from above, Figure 1a. The gates to the marginal rooms were shut for at least 8 min. For this part of the experiment, the rat’s head orientation had to be provided at every moment. The experiment went on by opening the two doors, so the free rodent could walk into the entire environment. Also, each of the two cages could be populated or not by another rat. For the final part of the experiment, the scene was viewed with a camera placed on one side of the plastic box, Figure 1b. The number of visits in each compartment and the duration of each visit had to be provided.

The experimental environment for the second application is presented in Figure 2. A 96-channel multielectrode array implanted in the dorsolateral prefrontal cortex (dlPFC; area 46 [23]) of the monkey acquired and transmitted the EEG signals via radio waves to be recorded and further analyzed by neuroscientists. The experiments conducted in the neuroscience lab of McGovern Medical School at the University of Texas are described in detail in [24]. Along with EEG signals, the sensor array position and monkey’s head orientation were also needed for an accurate final analysis. This task, one of the three applications presented in this paper, was performed using the registrations of the four experiments described earlier. The cage was unevenly illuminated. In addition, the monkey could switch the light at every moment. Besides the sensor array position and the monkey’s head orientation, an indicator for light-on/light-off was transmitted to be considered by specialists for final analysis. Figure 2a illustrates a light-on scene and Figure 2b depicts a light-off moment.

As for the last application, the scene was visualized by four video cameras, Figure 3. Each camera had a fixed position. There were two views of the entire arena viewed from two different angles (Figure 3a,b). The other two video cameras captured the background of the first two general scene views (Figure 3c,d). The habitat included six refuges. Only the refuge gates were visualized with the cameras. The bear might have quit the landscape entering one refuge. When the panda was in the domain viewed by the visitors, it could be visualized in at least one of the four views. Every time the zoo caretaker entered the cage to fetch food or to clean up the room, the bear was guided in a specific retreat. For the sequences with pandas in the field, five categories of stress episodes had to be identified. Each stress type was characterized by specific movement patterns.

The target localization using only the current frame could be accomplished by classic image object detection techniques only for the first part of the rat tracking and trajectory analysis application, Figure 1a. Due to the high contrast target/background, the rodent could be identified with a succession of single image operations: binarization -> erosion -> the main blob identification.

Head position, as explained in the next section, was determined using information from the frames in the vicinity of the current one. As for the rest of the presented applications, due to the low contrast target/background, a tracking technique had to be used. In this application, we faced great motion nonlinearity because of the big shifts in velocity and motion direction. In this respect, Kalman filter methods could not be employed. Also, due to brightness changes, tracking techniques issued from the optical flow technique could also not be used. Thus, the main supposition in the optical flow method was that the pixel intensities were translated from a certain frame to the next one: $I(\left(x,y,t\right)=I\left(x+u,y+u,t+1\right)$, where $I(\left(x,y,t\right)$ is the image intensity of the $\left(x,y\right)$ pixel at moment t, and $I\left(x+u,y+u,t+1\right)$ is the level of the pixel translated by the $\left(u,v\right)$ vector in the next frame at moment (t+1). From frame to frame, the target could perform significant movements from a darker area to a whiter zone. Largely used in the last years, movement tracking based on deep learning approaches was not suited in our case due to the fact that the computation was strongly scene-dependent for each of the three applications. For the first and the last systems, the parameters to be computed were related to the scene architecture. As for the monkey’s head orientation, the analysis was directed by the scene illumination specificities. 

We propose a localization method as follows:

For each consecutive three frames composed by the current frame, indicated by $I_{a}$, and the two anterior frames, denoted $I_{b}$ and $I_{c}$, respectively, perform:

$I_{dif-dif}=\left\lfloor I_{a}-I_{b}\right\rfloor -\left\lfloor I_{b}-I_{c}\right\rfloor$;$I_{mov}=D\left(E\left(I_{dif-dif}\right)\right)$, where *D* and *E* represent Dilation and Erosion morphological operations, respectively;In $I_{mov}$, delete all movement pixels external to the region of interest (this step is application-dependent);Through a ‘counting-box’ method, identify the most relevant movement cluster in $I_{mov}$. 

The procedure outcome is shown in Figure 4.

The movement cluster coarse localization provided by this procedure was refined using other information provided by the application-dependent procedures. For instance, in the case of the neuroscience experiment, to exactly identify the target location, the Canny image of the current frame was used. The right location was in the movement cluster mass center vicinity corresponding to the area in the Canny image where a hole made by the sensor box border was identified.

## 3. Rat Behavior Analysis

### 3.1. Head Localization

For the first part of the experiment, we proposed a three-stage method [19]: (1) for every frame, the scene characteristics were computed; (2) if the characteristics analysis indicates there were no scene outsiders, the rat body was identified; also, the rat head was recognized considering the head coordinates from the preceding frames. For doubtful cases, it was calculated in a second position; (3) the entire coordinates set was processed in order to solve each uncertain case considering subsequently certain head positions. 

The first step of the scene analysis is illustrated in Figure 5. A binarization threshold was computed so that for a normal frame, the analysis of the binary image projection on the vertical axis would detect five clusters delimited by $\left(s_{i},e_{i}\right),i=1\ldots 5$, parameters, Figure 5c. If the five cluster borders satisfied some predefined proximity conditions, the left and right middle room walls, $\left(b_{1},b_{2}\right)$, were in the middle of the second and fourth cluster, respectively. If the two parameters could be computed, the process continued by analyzing only the middle part of the cage in the current frame. Otherwise, the next frame would be analyzed. 

The analysis of a normal frame is depicted in Figure 6. A new binarization threshold was applied so that the main blob in the middle room could be easily located after an erosion of the binarized image. 

In the initial stage of the following session, because of the fact that the rat was positioned in the middle zone by the human operator, only vertical or horizontal positions were available. After the first two frames, analysis the two previous frame localizations strongly indicated the head position for the current frame. However, for the first two frames, the previous two head locations were not available, so a voting method involving two techniques was used for vertical or horizontal situations. Let us consider the horizontal case, which is depicted in Figure 6a. First, the vertical projections of a fraction of the left and right areas, respectively, of the horizontal framing rectangle were analyzed. The area with significantly fewer body pixels would contain the head. Secondly, in the binary image, the tiny areas outside the right part and outside the left part were scanned, Figure 6b, to detect the tail. The rat’s tail container was considered the area with a large amount of white pixels located in the middle. The decision was taken following these rules: (a) if both methods show the same head zone, the area mass center represents a head indicator, as illustrated in Figure 6e; (b) one of the two methods does not offer reliable information (the two vertical projections are almost similar or the tail could not be identified): The result comes from the other one if it provides a strong decision; (c) both algorithms provide strong but contradictory responses: The second one is validated; (d) if none of the first three conditions is satisfied, the decision is delayed for the next frames. If the previous two frames’ directions were available, the decision was taken considering the old direction. When the rat reached a box wall or turned around, the body framing quadrilateral was almost a square. In such cases, the head was localized based on the movement direction and head locations of the previous frames. If more movement directions were possible, the two most reliable results were considered. At the end of the tracking process, based on the entire list analysis, the uncertain cases (with two solutions) were solved by choosing the closest position to the location of the next frame with a reliable localization (one solution).

### 3.2. Compartment Visits Number and Visit Duration

To analyze low contrast target/background registrations of the final part of the experiment, we proposed a two-stage tracking method [20]: (1) a tracking step based on the frame differences approach presented in Section 2 to synthesize the trajectories of the moving rats and (2) trajectory offline analysis to compute the entry number of the free rat in each area (cages, middle room) and the visit durations.

The tracking process is depicted in Figure 7. The original frame with a free rat wandering through the middle room is shown in Figure 7a. As in the first part of the experiment, a binarization threshold was computed. This time, the computation was conducted so that for a normal frame, the analysis of the binary image projection on the vertical axis would detect the limits of the two cages on the left and the right of the whole box. If the result fulfilled the condition $0<c_{11}<c_{12}<c_{21}<c_{22},$ then compute $b_{1},b_{2}$ by analyzing the column projections in the interval $\left(c_{12},c_{21}\right)$. The image used for the experimental environment parameter computation is depicted in Figure 7b. Finally, if the deducted parameters satisfied the condition $0<c_{11}<c_{12}<b_{1}<b_{2}<c_{21}<c_{22}$, proceed to locate with a counting box approach the main movement cluster in the working image generated as explained in Section 2. For the frame in Figure 7a, the result is shown in Figure 7d. The working image generated by consecutive frame differences is shown in Figure 7c. At its bottom, the image also displays the cages and the plastic box extremities: $c_{11},c_{12},c_{21},c_{22},b_{1},b_{2}$.

For both stages of the rat behavior analysis project, the experimental box parameters had to be computed for each analyzed frame due to the fact the experiment conductor could interfere at any time during the process and might slightly change the scene angle view. 

### 3.3. Results and Conclusions

All of the two sorts of experiment registrations have been provided by our collaborators from the Center of Biomedical Research of the Romanian Academy, Iasi Branch. For the first part of the experiment, tests were performed on ten movies of various resolutions. The first two movies show only the application studied in our experiments. In the middle camera, the rat did not have the ability to go to the other two rooms. The other films last 30 min and start with an episode of about 8 min. The head identification is accurate for assessing the motion of the rat, and the body framing quadrilateral is a vertical or horizontal rectangle. We detected a few errors in the case of turn-around motions or motion direction changing cases. To decrease the errors in these cases, an entire position file processing was accomplished. One of the two possible places calculated for doubtful cases was selected by means of the proximity to the first certain position of the following frames. For each video, the number of frames with small positioning errors in the framing squares versus the total number of observed frames is detailed in [19]. If the overall analysis is not considered, the error rate varies between 2.4% and 3.5% except in the first recording in which the error rate is about 1.2%. This is because the freezing sequences of the monitored rats last longer than for the rest of this study. After the complete analysis, the error rate decreased by one-third for the recordings. 

As for the final part of the experiment, tests were conducted on three videos of different durations (10, 20, 40 min) and resolutions (788 × 246, 808 × 196, and 1092 × 270). The entry numbers in each area were determined errorless. The entry durations were computed error-free for short visits and with errors of a maximum of 1–2 s for visits longer than 1 min.

## 4. Animal Tracking Method Used in Neuroscience Experiments

### 4.1. Target Localization

As mentioned in Section 2, to locate the monkey in the current frame, a tracking technique was employed based on the identification of the main cluster in the image resulting from the processing sequences of the three consecutive frames (current frame and the two previous ones), Figure 8c. At the beginning of the tracking process, a model was yielded by storing the gray levels of the rectangle shape around the cluster center of gravity as soon as the following conditions were met: (a) the main detected cluster was relevant; (b) the Canny features, Figure 8b, of the current frame depicted in Figure 8a in the neighborhood of the movement cluster center of gravity were in a range specific for the Canny traits near the border of the empty patch yielded by the sensor box; (c) the average gray level of the cluster center of gravity neighborhood of the current frame indicated a dark area. 

The localization result is depicted in the working image, Figure 8d, and in the current frame, Figure 8d. Apparently, the model synthesis is very restrained because of the strong (a–c) conditions and the area borders where the model could be produced. Yet, the model initialization could take place only if the scene was illuminated. Now, the monkey moved a lot but also takes many breaks in the specific areas where the model could be initialized. 

After a model was produced, the watching conditions were simplified. A new position was deduced after the movement cluster localization was validated only on the basis of an acceptable model pattern matching score and Canny edges being in the right range. Yet, the model could be applied to consolidate the detection of the current position only if this one was in the close vicinity of the previous one. Every time a head position was calculated, the current model was updated. The pattern-matching approach had no significance when a very relevant motion cluster was found far away from the anterior position. Now, the tracking procedure started, and a new exemplary was produced if the conditions (a–c) were satisfied. 

Some precautions have to be considered. The entire process restarted in the case of the current frame being uniform. Another precaution is related to the unlighted frame sequences. The monkey shadow could yield false motion features. This problem can be approached by taking appropriate search zones. Moreover, it is important to choose the Canny traits number and their spatiality (in the sense that the close vicinity of the candidate center of gravity must be Canny features-free).

Four movement areas of the cage scene were considered. Each area has an experimentally determined reliability index. For example, the east domain of the cage has a smaller index when the monkey climbs the box wall, and its shape is not noticeable. The north domain of the cage has a low index, too. For every localization, a confidence index was calculated by means of the movement cluster area reliability index, possibly the model matching score, and the iteration number in case the recognition was resumed using more relaxed locations conditions. As another aspect of reliability index computing, the position calculated is assigned a lower confidence index if it is not computed in the first iteration, and the motion traits number is great. Ultimately, just localizations with an index bigger than 75% are useful for further assessment.

### 4.2. Head Orientation

Two stages are necessary to compute the head orientation. First, for each frame, the head orientation was calculated considering that the sensor set had a dark color, a right-angled profile, and was mounted in a determined position on the monkey’s head. Secondly, the orientation amounts were filtered, and a complete set of trends was computed.

Moreover, the area located in each frame by the following method was increased by enlarging its size in every direction by half of the primary measures. The new region of interest (ROI) resulting after inflation was converted into the HSV color space (hue, saturation, value). Then, a Gaussian filter was applied to the V component. 

The sensor array was identified in the new image with the following operations: (a) the image was binarized by means of a threshold amount in the domain [30; 60] according to the scene brightness; (b) all contours in the binary image were located; (c) for each contour with a significant area (more than 500 pixels), a filling index was calculated as a percentage between the area bordered by the profile and the surface of the smallest bounding rectangle; (d) the specific orientation was offered by the direction of the large part of the bounding rectangle of the area with the biggest filling index.

Finally, a 1-D average filter was used for the series of frame angles to clear the high-frequency noise and to estimate the angle measures related to the frames for which the orientation could not be calculated. The angle of the frame *i* calculated in the first step was changed by the mean of the angles in a temporal window of size *t* centered in *i*, from which the places where the angles were not calculated were eliminated, or the suitable confidence index was smaller than a certain threshold. A result of the target orientation computation is shown in Figure 9.

### 4.3. Results and Conclusions 

Four videos provided by McGovern Medical School at the University of Texas were used in the experiments. The movies’ resolutions are besides 917 × 688, and all of them have significant light off sequences. A large variation of illumination intensity for the lighted case is presented in the third film. Some of the registration sets contain frame sequences with no information. The angle of view is slightly modified from one video to another. 

The analysis outcomes are better detailed in [21]. The tracking error was defined as the percent of frames with a low localization confidence index where the ROIs were not taken into account in the orientation estimations. For three of the four recordings, the error is about 1%. Due to the issues shown in the previous paragraph, the computed error for the fourth sequence is 3.4%. The average angle correction during the filtering stage varies between 16.9° and 20.0°. These amounts are great but admissible if we know that for many frames, the primary direction and orientation cannot not be calculated because of the weak image’s quality.

Due to the fact that this application was deployed for specific experiments, it is hard to confront the results with those of other animal following developments. The experiment specificities consist of: 

-Unequal brightness for lighted scenes and also when light was banned; during the torn-off light episodes, the monkey shadow could yield forged motion traits;-The manner in which the experimental work was managed: Frequently the monkey's movement was tracked from the cage exterior, and the experimenter’s motion produced not useful movement traits; occasionally, the monkey was fed from the outside of the box, and this fact also does not generate monkey traits;-The abrupt shifts of the box brightness due to the fact the monkey could accidentally turn the light on/off;-The monkey motion is highly nonlinear. 

To conclude, the monkey localization and its head orientation tasks have an accuracy that makes this practice appropriate for this type of neuroscience experiment.

## 5. Stress Sequences Identification of a Zoo Panda Bear by Movement Image Analysis

The analysis input consists of sets of zoo films. A set can contain synchronized recordings from one to four video cameras. The four sights from Figure 3 are indicated with the camera number used in the experiment: c5-Figure 3d, c6-Figure 3b, c7-Figure 3a, and c8-Figure 3c. 

First, the application would initialize the following parameters needed via a tracking process and trajectory analysis:

-Motion domain selection: To cancel no information motion traits (e.g., visitors’ motions; flaring on walls or ceiling), for every view, a helpful information polygon was set up. Afterward, during the tracking stage, by means of a ‘ray casting’ method, just motion traits inside the polygon were analyzed;-Panda bear and zoological garden caregiver entry/exit gates coordinates in the information polygon for every sight;-The complete visibility zones were bordered for each sight. In these zones, it was possible that after a valid motion cluster recognition, there were no discovered motion traits. This could be motivated by a relaxation or freezing moment of a panda. A no-motion characteristic was set up to eliminate the tracking re-restoration;-For each visible area of a view, a list of the coordinates of the correspondent areas of the other cameras was synthesized.

Then, an already defined project could be launched, or a new project was defined: (a) the views to be fused from the available video set were chosen; (b) for each project view and for the final result, a list with an entry number equal to the minimum frame number of all the project video lengths was generated. Each list entry was a structure including a camera number, movement coordinates, and a movement attribute. For all lists, each list position was initialized to a null camera number, null movement coordinates, and a *freezing_or_missing* attribute. 

For every video camera comprised in the actual experiment, a tracking procedure was initialized. For at least two camera working sequences, these tasks were run in parallel. The next tasks were applied: (a) target localization for every camera session; (b) trajectory estimation from all trajectory views; and (c) obtain trajectory filtering and analysis. 

### 5.1. Localization at Task Level

To estimate the movement strength given by the number of features (features_nmb) of the localized movement cluster, three movement thresholds were used; they are denoted: features_nmb_thr, low_features_nmb_thr, and high_features_nmb_thr. A tracking session is initialized or reinitialized only if:

(a)features_nmb > high_features_nmb_thr. In this case, if the height movement rectangle is specific to the zoo technician and is seized in the zoo technician gate area, an attribute indicating technician presence is set: zoo_technician = 1. This parameter was used in the overall trajectory synthesis where only panda frames were considered;(b)features_nmb > features_nmb_thr, and the movement cluster is identified in the gate areas;(c)features_nmb > features_nmb_thr, and the movement cluster is identified in the previous valid position area. This case is possible only if the precedent tracking session provided valid results. 

A watching session could end after a caregiver or panda bear exits or after a long series of static frames.

During the tracking, according to features_nmb and the previous frame characteristic, in every sequence of the working session, the existing frame was assigned as being a movement or freezing or no_information frame. 

If features_nmb < low_features_nmb_thr and:

-In the preceding frame, the bear was detected in an observable zone, and the movement labeling the actual frame was marked as freezing, and the watching kept going; -The previous frame was labeled as freezing, and the current frame was also labeled as freezing;-Else, the frame was marked as no_information, and the watching was canceled. 

If features_nmb > features_nmb_thr, the picture was marked as a movement frame and was examined in order to be confirmed as one or not. 

If low_features_nmb_thr < features_nmb < features_nmb_thr and:

-The panda bear was viewed in the door area after an exit, the watching zone for the next video frame would be restrictive to the same door area. This specific seeking continued as long as the condition took place;-The bear was viewed in a visible area other than the door zone after a movement frame series, and the current motion rectangle was almost in the same position as the set of preceding rectangles, then the current frame was marked as freezing;-Else, the no_information attribute is set. 

When the panda was detected in a visible zone and the precedent frame was a movement frame, the motion-seeking area was confined around the current position. Otherwise, the searching area for the counting box method increased to the entire motion zone determined by the ‘ray casting’ method. A very specific watching case could be generated for films provided by camera number 8 when the panda bear had a bath merely behind the two foreground stones. The panda might have been invisible, but because of his bathing activity, the pond water started flooding the zone. In this way, a lot of motion traits were generated in the analyzed image. In this particular case, the watching took place in a limited zone beside the panda's last observable position.

Finally, if the primary was marked as a movement frame, the current frame was not selected as a candidate for trajectory synthesis if: (a) the movement rectangle area greatly surpassed the previous movement area. This could be generated by a bursting, huge movement (caused by flaring or abrupt brightness changes); (b) the motion rectangle was located far away from the previous positions in a visible area; (c) the motion rectangle position did not agree with the motion direction shown in the preceding motion. 

Another validation framework was generated by a sequence with the panda slowing down, sitting in a visible area, and starting to play with wood branches. In this case, the movement features were situated apart from the real panda position. For such sequences characterized by a fast movement followed by a stop in an observable area and subsequently far away motion rectangles, the tracking outcome is correlated with a frame assessment that identifies the panda corpus. If the image analysis indicates a panda location very near to the previous position, the current movement rectangle is ignored, and the previous validated movement rectangle is considered. 

The final step of the validation of the current frame analysis result consists of the correlation with the previous frame selected by the fusion process. In the case of cameras in the general view, c6 and c7, the watching result could be disturbed by flying birds. Now, the result was validated only if the movement features were located in an area corresponding to the zone detected in the previously detected frame. The previous selected camera could be different from the current one. To check the condition, the list of corresponding areas of the previously selected camera was used. 

The tracking task pseudo-code is presented in Appendix A.

### 5.2. Trajectory Synthesis

The frame with the most observable bear was chosen and archived in the final frame collection at the end of the current frame analysis for all the project cameras. The list would be processed in order to maximize the sequences of the same camera and finally would be analyzed to detect the stress episodes.

The frames with localizations in the exterior of the observable zone were not considered. The same held for those where the zoo technician is present.

As described in pseudo-code in Appendix B, the frame selection was applied by means of a gamut of rules that establish a hierarchy of observable zones for the four video cameras. 

### 5.3. Trajectory Analysis

The procedure of the trajectory synthesis provides the list of fused panda positions. Each entry of the final list also includes the frame number and the motion attribute (movement, freezing, or no_information). Also, the trajectories picked up from each project camera are available.

The synthesized trajectory was filtered in order to simplify the stress episode detection by generating more compact sequences of the same camera. The final trajectory synthesis performed at the end of each frame analysis chose the best available view. Sometimes, when the panda was moving in a full-view area, the target was sensed by at least two cameras. From the two options with almost the same information offered, the synthesis technique favors the one already selected previously. There are situations when the selection procedure switches to another available position because the previously selected camera did not provide a position due to some tracking error. Later on, the former selected camera could provide a better result than the currently selected view, and again, a switch occurs. 

For high-resolution registrations, the six filtering methods presented in [22] were improved. Due to the new implementation, the recognition rate increased for three specific stress categories. The six trajectory optimizations are linked to motion shapes: freezing–moving switches, frequent walks on the same scene path, etc. Each of the six filtering approaches employs the same technique but is applied to different contexts. So, we shall detail only the first filtering step, which is meant to detect the sequences bordered by large selection domains of the same camera. Inside the list, short position sequences of other video cameras are introduced and possibly alternate with a few video frames of the current camera. To obtain a more compact sequence of the main camera, each short succession of the complementary camera would be changed with the corresponding series of the current camera. The process is shown in Figure 10 where Figure 10a,d is the right end of the first large series of current cameras; Figure 10c,f is the left extremity of the second large series of current cameras. Figure 10b shows a selection substitution from the current camera to the complementary camera. The new camera series starting from the position shown in Figure 10b until the last position of the interleaved sequence would be changed with the correspondent sequence of a current camera whose start is shown in Figure 10e. The procedure is summarized also by the pseudo-code in lines 1–9. 

Finally, in order to identify the stress sequences, the following processing stages were implemented in the last variant of the computed path:

(1)For each frame gap $F_{i+j}-F_{i}\in \left\{2.100\right\}$ of the unfilled frame positions bounded by the same useful information from the camera, perform the following: Each parameter structure of gap intermediary frame $F_{i+k}$, k=1…j−1 would be filled with movement mass center coordinates: $x_{F_{i+k}}=x_{F_{i}}+k\frac{x_{F_{i+j}}-x_{F_{i}}}{F_{i+j}-F_{i}}$; $y_{F_{i+k}}=y_{F_{i}}+k\frac{y_{F_{i+j}}-y_{F_{i}}}{F_{i+j}-F_{i}}$, and the attribute would be labeled as moving if at least one pair position is marked as moving. Else, the new positions are marked as freezing;(2)For every element of the updated list filter, the motion characteristic by counting the number of every characteristic type in a window is centered in the current position. Ultimately, the position characteristic is the characteristic type with the utmost number of votes;(3)For every pair $\left(i,i+1\right)$ with an identical available camera number, assess the local velocity: $v_{i+1}=\sqrt{{\left(x_{i}-x_{i+1}\right)}^{2}+{\left(y_{i}-y_{i+1}\right)}^{2}}$. The velocity is then weighted by camera number and by camera resolution;(4)For every available place in the last list, filter the velocity by averaging the velocities of good places in a window centered in the actual position.

To assess the *stress sequences,* three thresholds were used: thr_v1>thr_v2>thr_v3. According to the velocity amount and the zoo surface of the examined frames, and for a particular velocity domain, there are defined five stress classes: stress running, stress walking 1, stress walking 2, stress walking 3, and stress stationary:

-Positions on the list with $v_{i}>thr\_v1$ are labeled as stress running;-Every place on the list, besides the stress feature, with ${thr\_v2<v}_{i}<thr\_v1$ and that is a partition of a moving series consequence of a recording period in a complete visible zone is marked as stress walking 1;-Every place on the list, besides stress feature, with ${thr\_v2<v}_{i}<thr\_v1$ and that is a partition of a series consequence of a still period in one of the garden housings is marked as stress walking 2;-Every place on the list, besides stress feature, with ${thr\_v3<v}_{i}<thr\_v2$ that is a partition of a series consequence of a still period in one of the garden housings, and the path series has at least one height, is marked as stress walking 3. Here, height is defined as the point located at a significant distance to the line approximating the path section;-Tag with the characteristic stress stationary: the frame series with a duration greater than 5 s in which the panda bear bathes in the basin and is viewed by camera 8. Every place in such a frame series must have the motion center of gravity situated close to the basin.

The start of a ‘stress running’ sequence is depicted in Figure 11a. The sequence end is illustrated in Figure 11b. The sequence is followed by a segment labeled as ‘stress walking 1′ whose start is depicted in Figure 11c and the end in Figure 11d. The first episode lasted for 1.5 s, and the second had a duration of 2.3 s. 

The start, the end, and two intermediary positions of a sequence labeled with ‘stress walking 2’ are illustrated in Figure 12. 

The start, the end, and two intermediary positions of a sequence labeled as ‘stress walking 3’ are illustrated in Figure 13.

The start and the end of a 30 s sequence of ‘stress stationary’ are illustrated in Figure 14.

Appendix C presents the pseudo-code of the list filtering and analysis procedure.

### 5.4. Results and Conclusions

This shown implementation was the theme of a feasibility research contract with the company SC Centric IT Solutions Romania and is related to the tracking and recognition of recorded animals’ actions or behaviors by video analysis.

The main window of the user interface is displayed below (Figure 15).

The methods were tested on a large set of video registrations of two to four synchronized recordings given by the beneficiary comprising 480 h of high-resolution and 150 h of low-resolution movies with zoo activity. 

Every tracking sequence was analyzed with great accuracy due to the correlation of the current position with the preceding frame information related to the last path obtained by the path fusion procedure. Actually, the computed trajectory at the final part of the tracking stage is errorless. Before the last step of filtering and analysis of the synthesized trajectory, due to the many camera switches, just classification of the running stress states could be readily accomplished because the running parts yielded long path series in the last list. The walking and stationary stress state recognition could be as easy as in the last positions list; they are constituted by series provided by the same video camera. The six optimization stages shown in the previous section led to a better recognition rate. 

The classification accuracy for the five stress classes varies between 90% and 96% in the case of the low-resolution recordings $\left(640\times 480\right),$ as it is detailed in [22]. As mentioned earlier, for the high-resolution recordings (1920×1080), the improvement of the filtering methods was conducted with a better recognition rate (Table 1) for three stress types from the whole stress set, namely *stress walking 2, stress walking 2,* and *stress walking 3.* Now, for the high-resolution videos, the classification rate for all stress classes varies between 96% and 98%. 

It is difficult to compare these recognition rates with other application results due to the specificities of the panda habitat that strongly affect the panda behavior. Three of the five stress patterns, namely *stress walking 2*, *stress walking 3*, and *stress stationary*, are directly related to the specific locations of the whole observed area. However, the results are good enough considering the final purpose of this application. 

Finally, the stress episode detection result is provided in an Excel format, as it is shown in the image below (Figure 16).

## 6. Conclusions

The paper describes three applications for tracking and studying the behavior of animals in captivity. The three experiments are different both in terms of the configuration and size of the scene in which the animals are located, as well as in the type of actions followed and analyzed. The applications were developed within research projects and contracts, and the results obtained allow the continuation of research for a more in-depth analysis.

## Figures and Tables

**Figure 1 sensors-23-07928-f001:**
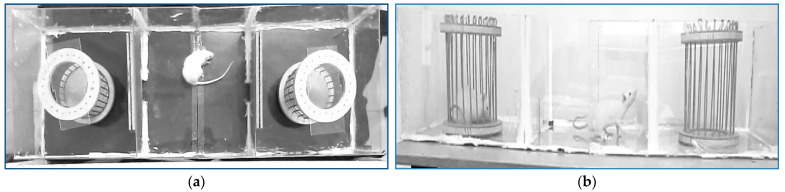
The two experimental environment configurations for rat tracking and trajectory analysis application: first stage (**a**) and second stage (**b**).

**Figure 2 sensors-23-07928-f002:**
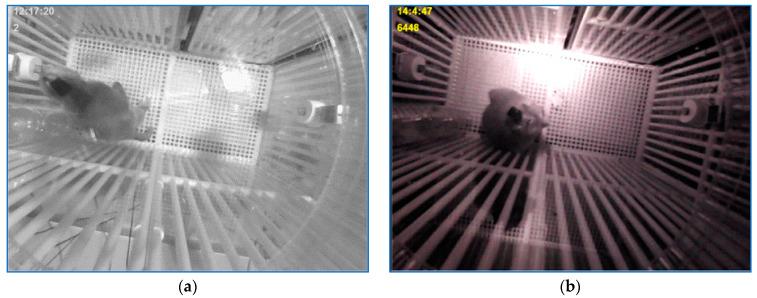
The laboratory environment for the neuroscience experiment: (**a**) lighted cage and (**b**) dark scene.

**Figure 3 sensors-23-07928-f003:**
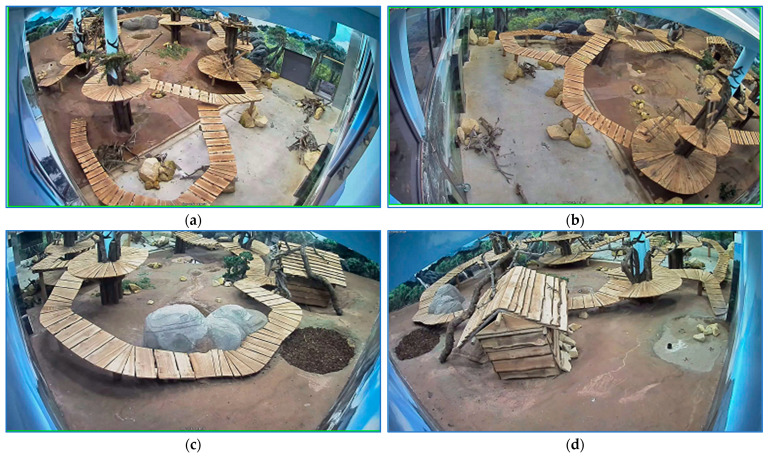
The arena is seen by 4 video cameras. Two cameras offer a view of the entire landscape viewed from two various angles (**a**,**b**). The other 2 cameras observe the ground of the general view (**c**,**d**).

**Figure 4 sensors-23-07928-f004:**
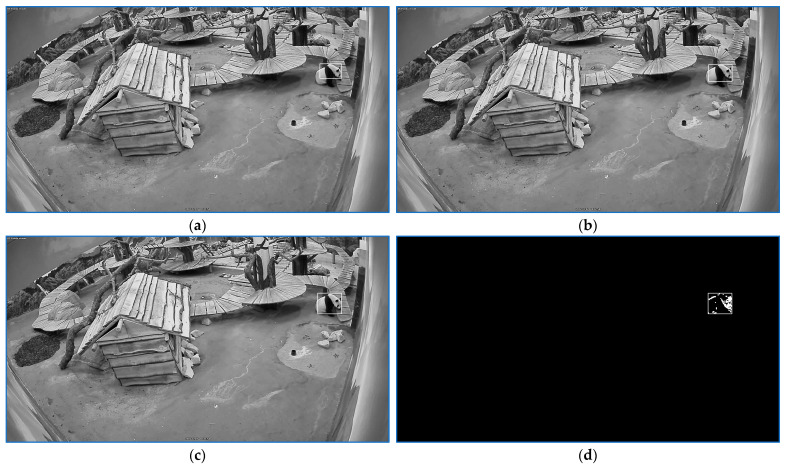
Motion traits generation. From the successive frame series, (**a**–**c**) generate target image shown in (**d**): $I_{mov}=D\left(E\left(\left\lfloor I_{a}-I_{b}\right\rfloor -\left\lfloor I_{b}-I_{c}\right\rfloor \right)\right)$.

**Figure 5 sensors-23-07928-f005:**
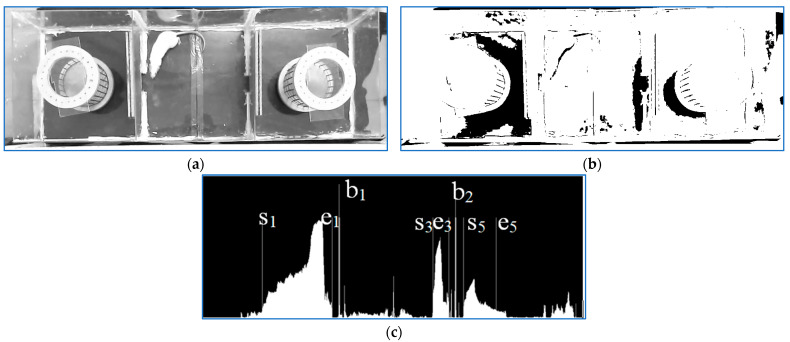
(**a**). Original frame; (**b**). binary frame; (**c**). binary frame column projection and the middle cage left $b_{1}$ and right $b_{2}$ walls.

**Figure 6 sensors-23-07928-f006:**
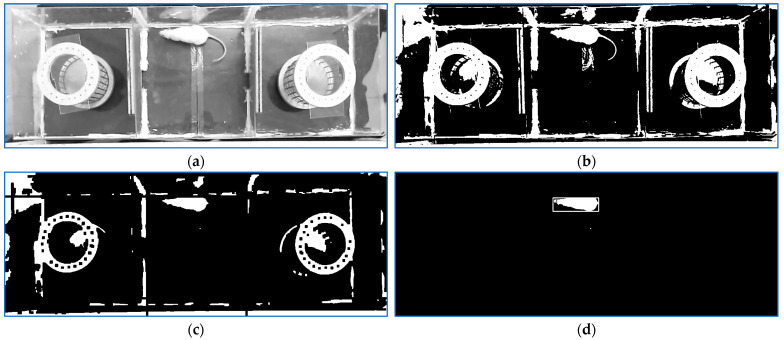
(**a**). Current frame; (**b**). binary frame; (**c**). erosion result; (**d**). rat and rat head localization results; (**e**). rat head detection result illustrated in the current frame.

**Figure 7 sensors-23-07928-f007:**
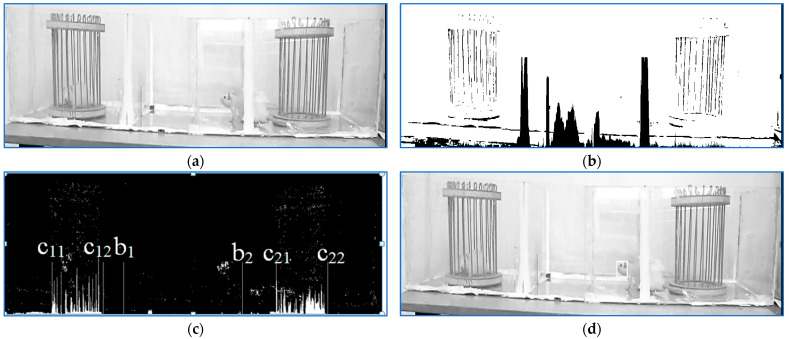
(**a**) Original frame; (**b**) generated image to compute the cage parameters; (**c**) working frame. At its bottom, the image also displays the cages and the plastic box extremities: $c_{11},c_{12},c_{21},c_{22},b_{1},b_{2}$; (**d**) tracking result.

**Figure 8 sensors-23-07928-f008:**
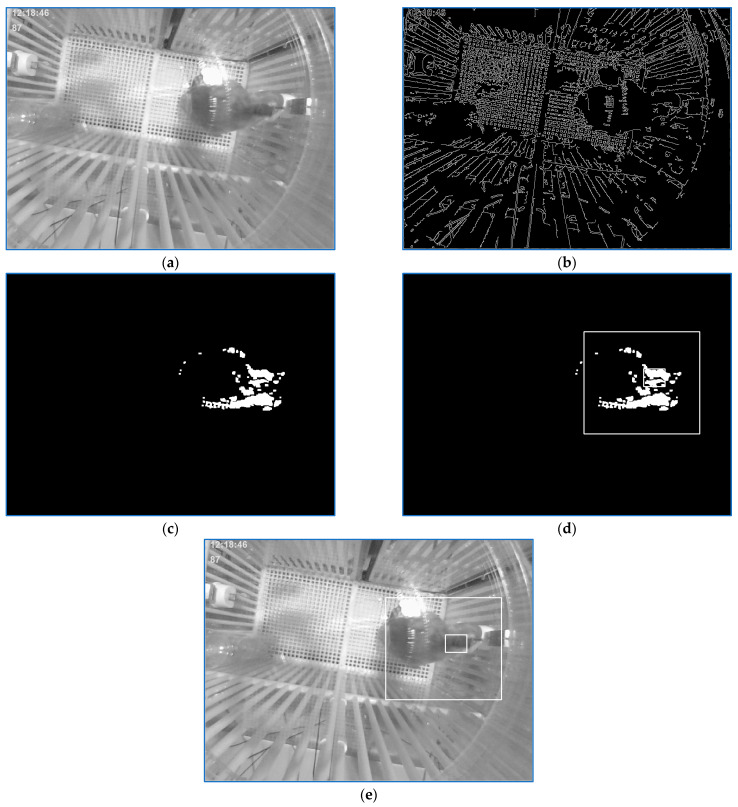
(**a**). Current frame; (**b**). canny features image of the current frame; (**c**). movement features image; (**d**). localization result provided by the movement features and Canny edges images analysis; (**e**). localization result visualized in current frame.

**Figure 9 sensors-23-07928-f009:**
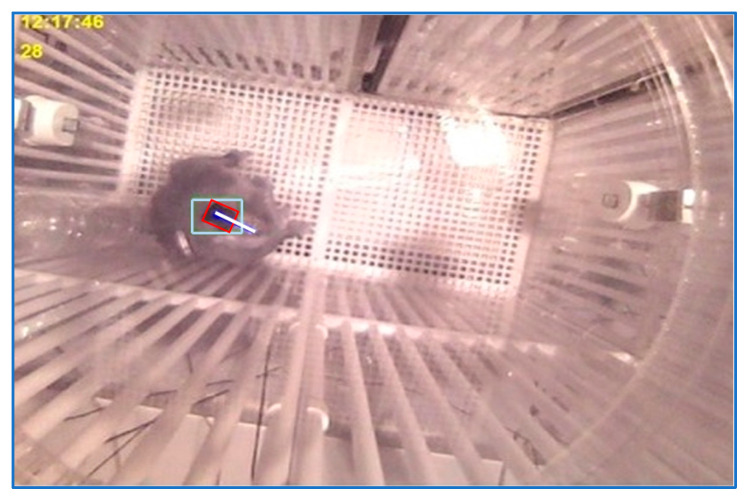
Head orientation calculus result. The red rectangle frames the sensors array antenna and the white line indicates the head orientation.

**Figure 10 sensors-23-07928-f010:**
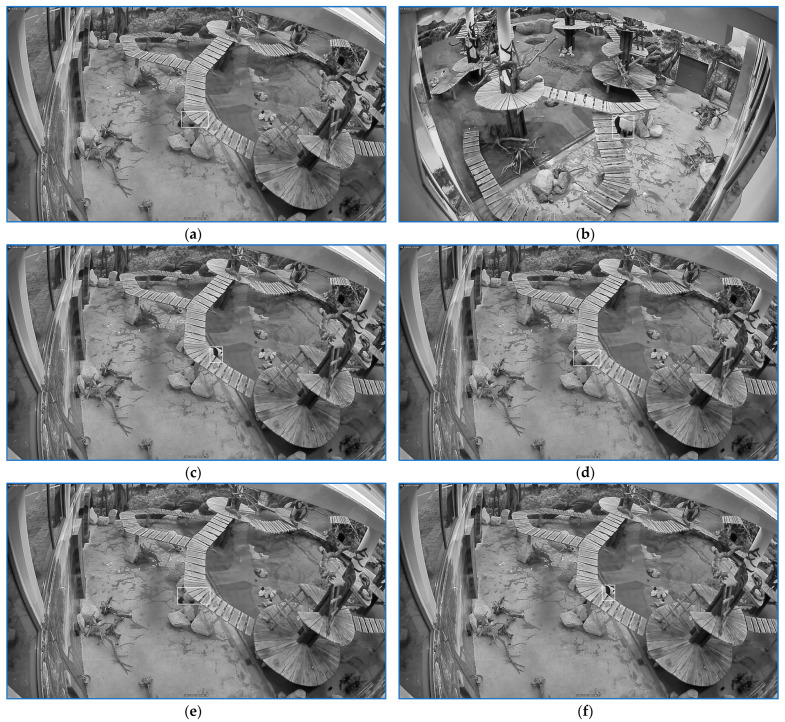
First filtering stage of the ending path. (**a**,**d**) depict the right end of the first sequence of current camera; (**c**,**f**) show the left extremity of the second sequence of the same camera. (**b**) shows the beginning of an in-terleaved sequence of the complementary camera. (**e**) indicates the beginning of the current camera sequence replacing the interleaved sequence of the complementary camera.

**Figure 11 sensors-23-07928-f011:**
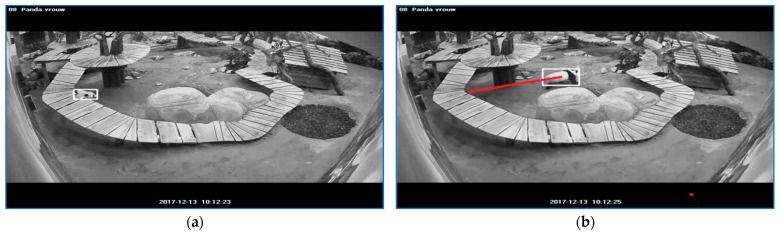
The start of a ‘stress running’ sequence is depicted in (**a**). The sequence end is illustrated in (**b**). The sequence is followed by a segment labeled as ‘stress walking 1’ whose start is depicted in (**c**) and the end in (**d**). The white rectangle frames the panda and the red line indicates the linear path that panda traveled for a specific stress sequence.

**Figure 12 sensors-23-07928-f012:**
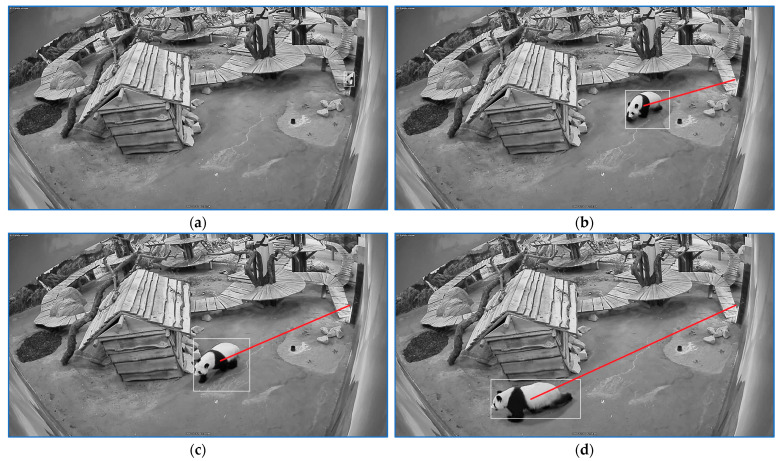
Four positions of a sequence labeled as ‘stress walking 2‘. (**a**) indicates the sequence start. (**d**) shows the sequence end and (**b**,**c**) depict intermediate positions on the path. The rectangle frames the panda and the red line indicates the linear path that panda traveled for a specific stress sequence.

**Figure 13 sensors-23-07928-f013:**
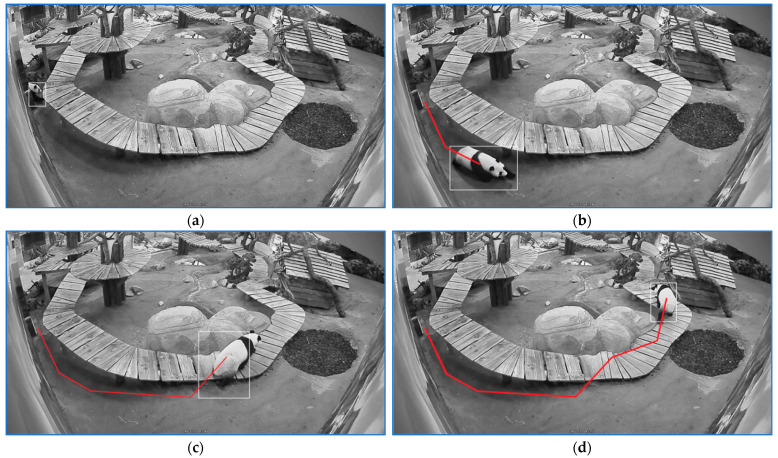
Four positions of a sequence labeled as ‘stress walking 3‘. (**a**) indicates the sequence start. (**d**) shows the sequence end and (**b**,**c**) depict intermediate positions on the path. The rectangle frames the panda and the red line shows the panda trajectory from the sequence start to the depicted position.

**Figure 14 sensors-23-07928-f014:**
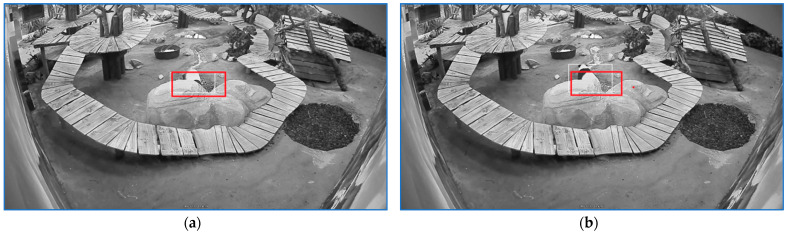
First frame (**a**) and last frame (**b**) of a sequence labeled as ‘stress stationary’. When panda is seized in the area behind of the two stones and the movement features cluster tends to elongate to the north the search area is restrained inside the zone depicted by the red rectangle.

**Figure 15 sensors-23-07928-f015:**
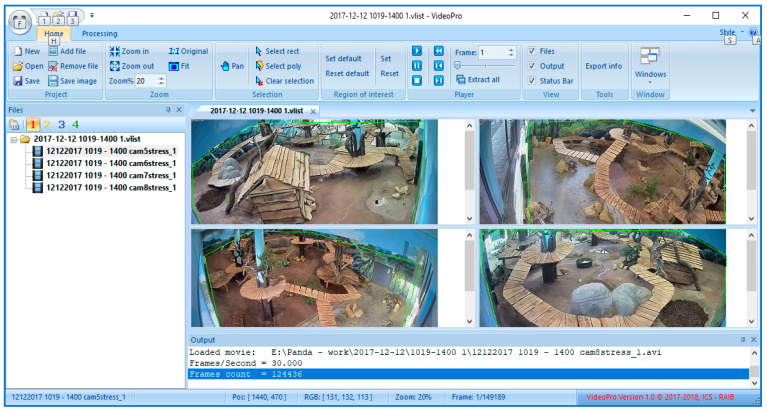
Main window of the application for animal activity recognition.

**Figure 16 sensors-23-07928-f016:**
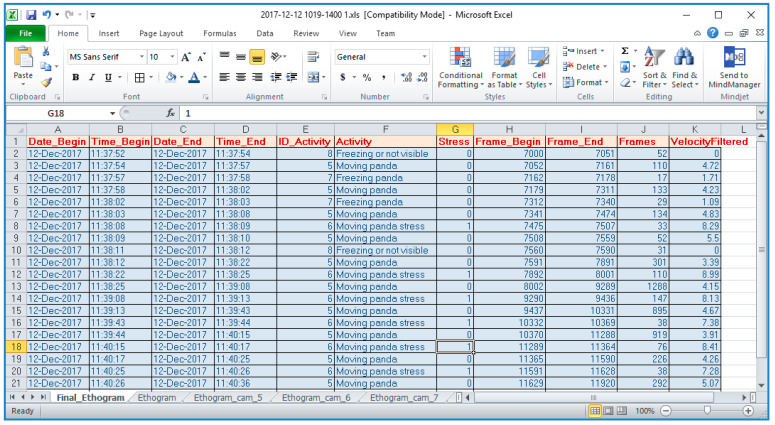
Stress episode detection. Processing results in Excel format.

**Table 1 sensors-23-07928-t001:** Improvement of stress episode rate recognition for high-resolution registrations.

	Score		
	Walking 1	Walking 2	Walking 3
Before improvement	94%	97%	95%
After improvement	96%	98%	97%

## Data Availability

Restrictions apply to the availability of this data. Data were obtained from application beneficiaries for this research only.

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
