# Peer review of "Captive Animal Behavior Study by Video Analysis"

_sensors, 2023, doi:10.3390/s23187928_

Round 1

Reviewer 1 Report

In this paper the authors presented three video analysis-based applications for captive animal behavior. Although the topic is interesting, they are some shortcomings, which are given below:

The role of EEG should be given in more detail. Did authors record EEG signals or electrocorticography (ECoG)? How many electrodes were used etc?

The conclusion section given in more detail. The authors should discuss the results of experiments in detail.

How can authors generalize the results? In order to come a firm decision, it should be better to apply more experiments with more number of rats, panda and monkey.

The Figures can not be seen well enough. Their resolutions and sizes should be edited.

How did the authors determine the parameters in analysis phase.

The authors should express the differences between this present and previous studies. Their previous study has a reference number [23] in the manuscript.

The English of the paper well.

Author Response

For research article

Captive Animal Behavior Study by Video Analysis

Response to Reviewer 1 Comments

Thank you very much for taking the time to review this manuscript. Please find the detailed responses below and the corresponding corrections highlighted.

Comments 1: The role of EEG should be given in more detail. Did authors record EEG signals or electrocorticography (ECoG)? How manyelectrodes were used etc?

Response 1: The recordings were made and provided to us by the McGovern Medical School at University of Texas, as is detailed in lines 155-166 of the paper.

Comments 2: The conclusion section given in more detail. The authors should discuss the results of experiments in detail

Response 2: We enhanced the conclusions section for the third application, in the final stage of implementation. Lines 644-651 in the paper.

Comments 3: How can authors generalize the results? In order to come a firm decision, it should be better to apply more experiments with morenumber of rats, panda and monkey

Response 3: The applications were implemented for very specific experiment environments. The tracking method is used for all three applications and could be adapted for other kind of experiments, but the final delivered parameters are very application specific.

Comments 4: The Figures cannot be seen well enough. Their resolutions and sizes should be edited

Response 4: All figures were resized.

Comments 5: How did the authors determine the parameters in analysis phase

Response 5: All the parameters were experimentally determined.

Comments 6: The authors should express the differences between this presen tand previous studies. Their previous study has a reference number [23] in the manuscript.

Response 6: The improvements are highlighted in the conclusions section, lines 644-651 in the paper.

Reviewer 2 Report

I appreciate the effort of the authors to create an interesting article.
I propose to publish it in the form presented

Author Response

For research article

Captive Animal Behavior Study by Video Analysis

Response to Reviewer 2 Comments

Thank you very much for your evaluation.

Reviewer 3 Report

This article introduces three video analysis-based applications for studying the behavior of captive animals. The first application aims to provide parameters for evaluating drug effects by analyzing the movement of rats, including indicators such as the head posture of rats, the number of times they enter each area, and their duration of stay. The second application monitors the position and head direction of laboratory monkeys to provide relevant information for neuroscience experiments. The third application demonstrates how to analyze the displacement and movement of pandas in a zoo through video analysis to identify their stress state.

The  strengths:

  1. The author proposes a universal localization method that uses the differences between three consecutive frames to identify moving targets, which is suitable for low-contrast and nonlinear motion scenes.
  2. The author designs different feature extraction and trajectory analysis methods for different application scenarios, demonstrating high accuracy and reliability.

The weaknesses :

  1. Lighting changes are an important factor that affects animal behavior, but the author did not discuss lighting in the experiments.
  2. The author did not discuss the limitations or scope of the method.

Author Response

For research article

Captive Animal Behavior Study by Video Analysis

Response to Reviewer 3 Comments

Thank you very much for taking the time to review this manuscript. Please find the detailed responses below and the corresponding corrections highlighted.

Comments 1: Lighting changes are an important factor that affects animal behavior, but the author did not discuss lighting in the experiments.

Response 1: We mentioned that besides the head orientation and position the status of the light on/off indicator was provided to the neuro-scientists to eventually improve their final analysis.

Comments 2: The author did not discuss the limitations or scope of the method.

Response 2: The applications were implemented on demand for specific experiment environments. The tracking method is used for all three applications and could be adapted for other kind of experiments that involve tracking methods, but the final delivered parameters depend on the application.

Round 2

Reviewer 1 Report

The authors addressed all the comments. I think this version is acceptable.